# Factual Relation Discrimination for Factuality-oriented Abstractive Summarization

**Zhiguang Gao[1], Peifeng Li[1], Feng Jiang[2,3,4], Xiaomin Chu[1], Qiaoming Zhu[1]**

[1]School of Computer Science and Technology, Soochow University, Suzhou, China
[2]School of Data Science, The Chinese University of Hong Kong, Shenzhen, China
[3]Shenzhen Research Institute of Big Data, Shenzhen, China
[4] University of Science and Technology of China, Hefei, China
1503022536@qq.com, pfli@suda.edu.cn, jeffreyjiang@cuhk.edu.cn
{xmchu, qmzhu}@suda.edu.cn

## Abstract

Most neural abstractive summarization models are capable of producing high-quality summaries. However, they still frequently contain factual errors. Existing factuality-oriented abstractive summarization models only consider the integration of factual information and ignore the causes of factual errors. To address this issue, we propose a factuality-oriented abstractive summarization model DASum, which is based on a new task factual relation discrimination that is able to identify the causes of factual errors. First, we use data augmentation methods to construct counterfactual summaries (i.e., negative samples), and build a factual summarization dataset. Then, we propose the factual relation discrimination task, which determines the factuality of the dependency relations in summaries during summary generation and guides our DASum to generate factual relations, thereby improving the factuality of summaries. Experimental results on the CNN/DM and XSUM datasets show that our DASum outperforms several state-of-the-art benchmarks in terms of the factual metrics.

## 1 Introduction

Using pre-trained models, neural abstractive summarization (Lewis et al., 2020; Zhang et al., 2020) can produce fluent and informative summaries. However, the facts in the generated summaries are often inconsistent with the source documents. For example, Cao et al. (2021) and Goyal and Durrett (2021) show that nearly 30% of the summaries generated by recent neural models have factual errors. Thus, factuality-oriented abstractive summarization is intended to improve the factuality of summaries and make summarization models applicable in practice.

Recent work on improving the factuality of generated summaries relies heavily on extracting facts from the source documents and integrating these facts into abstractive summarization models. These approaches can partially deepen the model's perception of facts. However, they still cannot find the cause of factual errors. One of the reasons for this is that the available summarization corpora contain only factual summaries and ignore counterfactual summaries. If a counterfactual summary has the annotated causes, it will help the model to identify the causes of the factual errors and then improve the performance of factuality-oriented abstractive summarization. For example, there is a document whose summary is "Tom stole the cheese". If its counterfactual summary is "Jerry stole the cheese" and the annotated cause is entity swapping (i.e., replacing "Tom" with "Jerry"), it is possible to train a model to deal with different factual errors based on their causes.

In this paper, we introduce a factual relation discrimination task to factuality-oriented abstractive summarization, which can make the model aware of the causes of factual errors and learn to avoid factual errors during the summary generation process. Specifically, we propose a DASum model based on factual relation discrimination to study factuality-oriented abstractive summarization.

We first use data augmentation methods (e.g., pronoun swapping, sentence negation, time and date entity swapping, quantifier swapping, and named entity swapping) to construct counterfactual summaries, which, together with the annotated factual summaries, form the factual summarization dataset. Experimental results on the CNN/DM and XSUM datasets show that our DASum outperforms several state-of-the-art benchmarks on the factual mtrics (i.e., FactC, DAE and SENT). We summarize our contributions as follows:

- We use a series of data augmentation methods to construct the factual summarization dataset to simulate the distribution of factual errors in the model-generated summaries;

- We propose the factual relation discrimination task, which determines the correctness of the

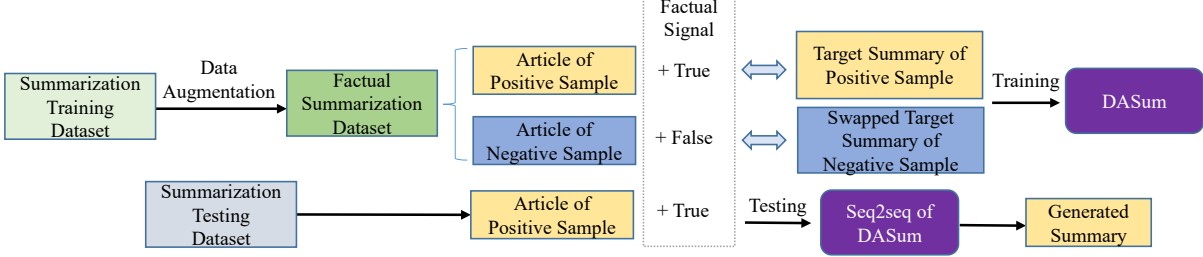

Figure 1: Overall process of our model DASum.

facts in the summary when generating, guiding the model to generate correct factual relationships, and thereby enhance the factuality of summaries.

## 2 Related Work

Previous studies on improving the factuality of summaries mainly fall into two categories, i.e., fact integration and reinforcement learning.

Fact integration is based on integrating factual knowledge into the summarization model. Cao et al. (2018) extract relational triples from the source document as internal facts, and use a dual encoder to encode the source document and internal facts simultaneously. Zhu et al. (2021) also take relation triples as internal facts and build the relation triple graph by connecting different triples. They then encode the relation triples graph with a graph neural network and feed the graph representation into the decoder of a seq2seq model. Gao et al. (2022b) encode internal facts and external Knowledge Graph information simultaneously to alleviate internal and external factual errors. Gao et al. (2022a) analyse the fine-grained categories of factual errors and create fact graphs to represent the factual information in the source document and target summary. They then employ adversarial learning to integrate the fact graph into the summarization model. Balachandran et al. (2022) propose a factual error correction model, which improves the factuality by modifying summaries.

Recent studies (Choubey et al., 2021; Cao et al., 2021) also improve the factuality of summaries through reinforcement learning (RL). Choubey et al. (2021) use two factuality metrics, entity overlap metric and dependency arc entailment metric, to compute the factuality score and then use it as a reward for RL. Cao et al. (2021) focus on external factual errors and annotate the external factual dataset XENT, a factual discriminator is trained on XENT, and the discriminator score is used as a reward for RL. However, because the accuracy of the factual evaluation metric is not good enough, RL doesn't really make the model understand the facts in the source document.

## 3 Methodology

Fig. 1 shows the overall process of our model DASum. In the training phase, the factual summarization dataset is first constructed through data augmentation strategies, and then the DASum model is trained together with factual signals; in the testing phase, the correct factual signal "true" is added to all samples to generate summaries.

Fig. 2 shows the architecture of DASum. DASum has different training modes for positive and negative samples on the factual summarization dataset. For positive samples, as shown in Fig. 2(b), the correct factual signal (True) is added to the end of the source document. At this point, all factual relations in the summary are factually correct; for negative samples, as shown in Fig. 2(a), the counterfactual signal (False) is added, and there are both factual and counterfactual relations in the summary.

In this section, we first introduce the construction process of the factual summarization dataset, then explore the representation methods of core words and dependent words in factual relations, and finally elaborate on the factual relation discrimination.

### 3.1 Construction of Factual Summarization Dataset

The main effort of previous research to enhance the factuality of generated summaries is to better integrate factual information into the model, ignoring the causes of factual errors. To address this issue, we train both the abstractive summarization task and the factual relation discrimination task using the factual summarization dataset on both factual and counterfactual summaries.

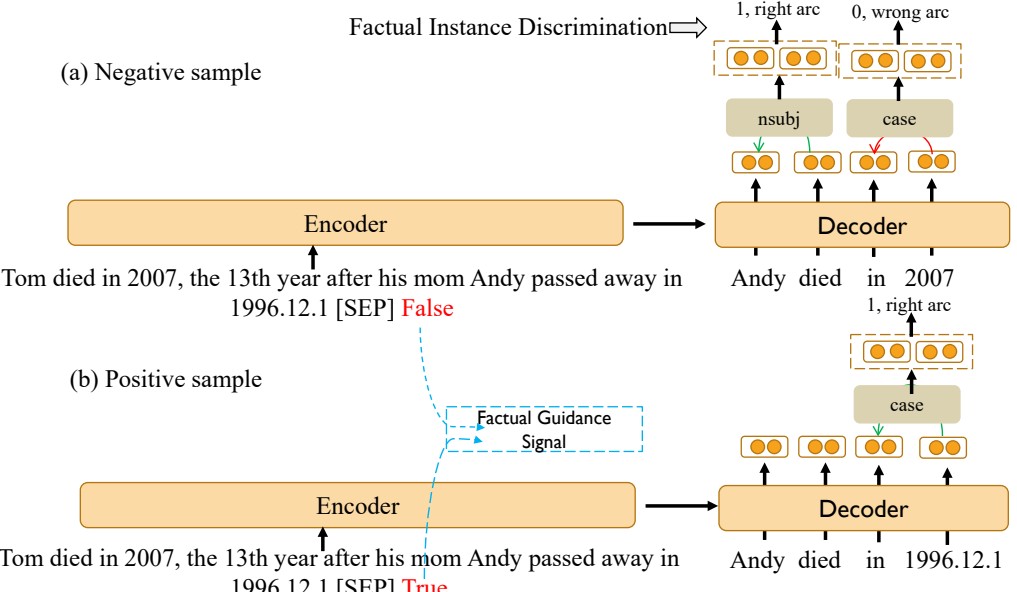

Figure 2: The architecture of DASum.

The factual summarization dataset not only contains factual summaries (positive samples), but also contains counterfactual summaries (negative samples). It is oriented from the existing summarization datasets (e.g., CNN/DM and XSUM) which only have factual summaries. In this paper, we use various kinds of dependency relation-based transformation rules to construct counterfactual summaries. That is, by modifying the dependency words of some dependency relations in a summary, the modified summary is counterfactual. For example, the time in a factual summary "Tom died in 2023.6.20" is modified to "2007", we can obtain a counterfactual summary "Tom died in 2007". The negative samples are constructed as follows.

First, we divide the training set of the original summarization dataset into two parts PO and NE, where the original summaries in PO are positive samples of the new factual summarization dataset and its negative samples are constructed based on NE by transformation rules. The discussion of different portion of PO and NE can be found in Subsection 4.4.

Then, we use the Spacy (Altinok, 2021) tool to obtain the dependency relations from each summary in NE. A dependency relation is an arc that points from the core word to the dependent word. According to the analysis on the causes of the factual errors in previous work (Gao et al., 2022b), we remove the relations whose dependency words are not one of pronouns, auxiliary verbs, time, dates, quantifiers, or named entities. Here, we simply use the operation of dependent word swapping to construct negative samples.

Since a summary may contain more than one dependency relation, we can transfer a factual summary to many counterfactual summaries by modifying one or more dependency words. To balance the rate of positive and negative samples, we only construct one counterfactual summary for each summary in NE. Particularly, only a part of dependency relations are used to construct the negative samples. That is, only the selected pronouns, auxiliary verbs, time and date entities, quantifiers, and named entities are transformed, while the rest are kept in their original state. This can train the model to discover the reasons for factual errors, rather than coarsely discriminating the factuality of the entire summary. The transformation rules for dependent words in the summary are as follows.

**Pronoun swapping** replaces pronouns in the summary with other pronouns of the same type. Pronouns refer to words that have the function of replacing or indicating people or objects, and they can be divided into subject pronouns, object pronouns, possessive pronouns, and reflexive pronouns. Gao et al. (2022a) classify factual errors at a fine-grained level and point out that coreference errors are a common type of factual errors in the generated summaries. The cause of coreference errors is mainly due to the ambiguity and confusion of the model regarding the objects referred to by pronouns. To solve this problem, we replace the pronouns in the summary to deepen the model's

connection to the pronouns and the objects they refer to. Specifically, for pronouns that exist in the summary, we replace them with other pronouns of the same type. An example is shown as follows.

E1: **Original summary sentence:** Her sister, shaneah, was dating lloyd.

**Modified summary sentence:** His sister, shaneah, was dating lloyd.

In E1, the pronoun "her" in the original summary sentence belongs to the possessive pronoun and is a denendency word. We randomly select another possessive pronoun (e.g., his) to replace "her". It is worth noting that some pronouns have multiple types, such as "her", which is both an object pronoun and a possessive pronoun. Since part of speech parsing tools are unable to further classify pronoun types, we randomly assign a type to pronouns with multiple types, such as calling "her" a possessive pronoun.

**Sentence negation** is a reversal of the semantics expressed in the summary, with the positive state changing to the negative state or the negative state changing to the positive state. When doing sentence negation, we focus on the auxiliary verbs in English (e.g., are, is, was, were). The summary generated by the abstractive summarization model sometimes confuses the negative form of sentences, for example, the negative statement that appears in the source document while the content in the summary is a positive statement. To solve this problem, we deepen the model's understanding of the positive and negative states of sentences by changing the negative form of the auxiliary verbs. Specifically, if the summary sentence has an auxiliary word, changing the negative form of the sentence can be divided into the following three steps:

(i) We determine the positive or negative status of the summary sentence. If the auxiliary verb is followed by "not", "n't", or "not", the summary sentence is in a negative state. Otherwise, it is in an positive state;

(ii) If the state of the summary sentence is positive, we add the negative words "not", "n't", or "not" after the auxiliary verb;

(iii) If the state of the summary sentence is negative, we remove the negative word after the auxiliary verb.

E2 shows an example of sentence negation. The summary sentence has the auxiliary verb "may" and is in an positive state. Then, the sentence is changed to a negative state by adding " not" after

"may".

E2: **Oringinal summary sentence:** Peter may have enough to spread around.

**Modified summary sentence:** Peter may not have enough to spread around.

**Time and date entity, quantifier, and named entity swapping** respectively refer to replacing time and date entities, quantifiers, and named entities that exist in the summary with other different entities of the same type. The cause of factual errors in generated summaries is usually due to the model's hallucination with the entity objects in the source document. To alleviate this situation, we replace the time and date entities, quantifiers, and named entities that exist in the summary with other entities of the same type to train the model to identify whether the generated entities are correct. First, we use the Spacy (Altinok, 2021) tool to identify entity types and select representative time and date entities, quantifiers, and named entities in the source document and the gold summary. Then, we replace the entities presenting in the summary with entities of the same type in the source document, and an example is shown as follows.

E3: **Original summary sentence:** Andy died in 1996.12.1.

**Set of DATA type entities in the source document:** $\{1996.12.1, 2007, this week, Friday\}$

**Modified summary sentence:** Andy died in 2007.

In E3, "1996.12.1" is an entity of the type DATE. In this case, we use other entities of the type DATA, such as "2007" selected from the source document, to replace "1996.12.1", thus forming a factual sample with factual errors. Specifically, we do not directly select an entity from the summary, mainly because the content of the summary is not as rich as the source document. By selecting an entity from the source document, the model can be further trained to discriminate the correctness of factual samples.

Finally, when the rate of PO to NE is set to 1:1, the positive and negative samples in the factual summarization dataset are 143,223 and 143,890 based on the CNN/DM's training set, 102,018 and 101,999 based on the XSUM's training set. Besides, our task of factual relation discrimination is to judge whether a dependency relation is true or false in the corresponding summary and we also must construct a training set to learn a model to discriminate factual and counterfactual samples.

Hence, we randomly select 10 factual dependency relations and no more than 5 counterfactual relations from a summary in the CNN/DM dataset to construct the training set of factual relation discrimination. For the XSUM dataset, these figures are 4 and 2, since the summaries in CNN/DM is long than those in XSUM. Finally, there are 1,831,627 positive relations and 640,541 negative relations in the training set of the factual relation discrimination task on CNN/DM , while these figures are 653,012 and 214,647 on XSUM.

## 3.2 Representation of Factual Relations

When constructing the factual summarization dataset, each dependency relation has two core elements: the dependent word and core word. The dependent word and core word are composed of words or phrases and can be considered a span of text. This subsection describes the representation of the span text.

When the summary is decoded through the seq2seq (Lewis et al., 2020) model, it is tokenized to obtain its token sequence. We extract the token sequences $D = \{d_1, d_2, \cdots, d_t\}$ and $H = \{h_1, h_2, \cdots, h_k\}$ corresponding to the dependent words and core words, where $t$ and $k$ refer to the number of tokens in $D, H$. The representation of dependent words and core words can be calculated as follows:

(i) **Boundary Embedding.** In this paper, we use BART (Lewis et al., 2020) as the basic architecture of the abstractive summarization model, and the output of the last layer of the hidden layers of the BART decoder will be used as the feature representation of the summary. The feature representation corresponding to the token sequence of the dependent word $D$ is $D_f = \{f_{d_1}, f_{d_2}, \cdots, f_{d_t}\}$. We concatenate the first and last token features of the dependent word as the final feature representation of the dependent word, i.e. $z_d = [f_{d_1}; f_{d_t}]$. We use the same representation for the core word yields $z_h = [f_{h_1}; f_{h_k}]$.

(ii) **Length Embedding.** We introduce length embedding (Fu et al., 2021) to expand the feature representation of dependent and core words. Specifically, we will first establish a length feature query table and then query the length feature table based on the length of dependent words and core words to obtain the length vectors $z^t, z^k$ of dependent words and core words.

The final feature representations of dependent

words and core words are $s_d = [z_d; z^t]$ and $s_h = [z_h; z^k]$, respectively.

## 3.3 Factual Relation Discrimination

DASum uses the task of factual relation discrimination to guide in mining the causes of factual errors. As we mentioned in Subsection 3.1, there are multiple factual relations $S = \{(D_1, H_1), \cdots, (D_m, H_m)\}$ in a summary $SUM$. Each factual relation is a dependency relation, where $D_i$ is a dependent word and $H_i$ is a core word. Factual relation discrimination is to determine the correctness of a given dependency relation in the generated summary.

For factual relation $(D_i, H_i)$, we first obtain the feature representation $s_d^i = [z_d; z^t], s_h^i = [z_h; z^k]$ of the dependent and core word through section 3.2. Then, we splice $s_d^i, s_h^i$ to obtain the representation of the factual relation $s^i = [s_d^i; s_h^i]$. Finally, we do the factual binary classification of factual relations through linear regression, as shown in Eq. 1, where $\sigma$ is the sigmoid activation function, $y_t$ is the factual label, and $W, b$ are the parameters to learn.

$$
\begin{aligned}
logits &= \sigma(W s^i + b) \\
L_{fact} &= -\frac{1}{m} \sum_{t=1}^{m} y_t log(logits_t)
\end{aligned}
\tag{1}
$$

Considering that the factual relation discrimination task is to assist the abstractive summarization model in better generating summary, we add the contrastive language loss $L_{lang}$, as shown in Eq. 2, where $\theta$ is the parameter of the seq2seq model, $Positive$ and $Negative$ are the positive and negative sets constructed in Subsection 3.1, and $X, Y$ represent the source document and summary, respectively. $L_{lang}$ is based on the idea that for a positive instance, the model should minimize its MLE loss, while for a negative instance, since the negative instance has undergone phrase replacement, to prevent the language model from learning the wrong posterior probability, it is necessary to maximize its MLE loss.

$$
\begin{aligned}
L_{lang} &= \mathcal{L}_{con}^+ + \mathcal{L}_{con}^- \\
\mathcal{L}_{con}^+ &= -\mathbb{E}_{X,Y \in Positive} log(p_\theta(Y|X)) \\
\mathcal{L}_{con}^- &= -\mathbb{E}_{X,Y \in Negative} log(1 - p_\theta(Y|X))
\end{aligned}
\tag{2}
$$

The final loss is as follow.

$$
L = L_{fact} + L_{lang}
\tag{3}
$$

### 3.4 Factual Signal

DASum is trained to understand the causes of factual errors by factual relation discrimination. However, DASum is not explicitly guided to generate factual summaries during the validation process. To address this issue, we add a factual signal to control the direction of model-generated content. Specifically, for source document $X = \{x_1, \cdots, x_i\}$, after adding factual signal, it becomes $X = \{x_1, \cdots, x_i, sep, flag\}$, where $flag$ is a factual signal. During the training process, when the instance is positive, $flag = True$; otherwise, $flag = False$; During the validation process, $flag$ is always $True$.

## 4 Experimentation

In this section, we first introduce the experimental settings, and then report the experimental results. Finally, we give the ablation study and analysis.

### 4.1 Experimental Settings

Following previous work (Pagnoni et al., 2021; Nan et al., 2021), we evaluate our DASum on the popular CNN/DM (Hermann et al., 2015) and XSUM (Narayan et al., 2018) datasets, and employ ROUGE (Lin, 2004) to evaluate the informativeness of the summary and the automatic factual evaluation metric: DAE (Goyal and Durrett, 2021), SENT (Goyal and Durrett, 2021) and FactC (Kryscinski et al., 2020) to evaluate the factuality.

In the data processing phase, we set the maximum number of tokens for the source document and gold summary of CNN/DM to 1024 and 200, and 1024 and 80 for XSUM, to deal with the inconsistency in the distribution of the summary lengths of the two datasets. In the training phase, in order to fully utilize the advantages of abstractive summarization pre-training, we use facebook/bart-large-cnn and facebook/bart-large-xsum as the pre-training parameter for the DASum model on the CNN/DM dataset and XSUM dataset, respectively.

### 4.2 Experimental Results

To verify the effectiveness of our DASum, we conduct the following baselines.

1) **BERTSum** (Liu and Lapata, 2019) takes BERT as the encoder and initializes the parameters of the decoder randomly.

2) **Unilm** (Dong et al., 2019) employs special mask strategies for pre-training and is applicable to both NLU and NLG task.

| Model | R1 | R2 | RL | DAE | SENT | FactC |
|---|---|---|---|---|---|---|
| CNN/DM | | | | | | |
| BertSum | 41.43 | 19.05 | 38.55 | 71.74 | 87.51 | 53.93 |
| Unilm | 43.33 | 20.21 | 40.51 | 65.21 | 82.32 | 36.43 |
| FASum | 40.53 | 17.84 | 37.40 | 73.57 | 79.12 | 50.14 |
| FASum$_{FC}$ | 40.38 | 17.67 | 37.23 | 73.80 | 80.36 | 51.17 |
| BART | 43.86 | 20.92 | 40.64 | 72.75 | 84.94 | 49.60 |
| CONSEQ | 44.40 | 21.37 | 41.17 | - | - | 72.83 |
| CLIFF | - | - | 41.01 | - | - | 50.05 |
| FACTEDIT | 42.17 | 20.22 | 39.37 | 75.71 | - | 75.49 |
| LASum | 43.25 | 20.21 | 40.09 | 82.51 | 87.61 | 82.40 |
| DASum | 44.27 | 21.18 | 40.99 | 77.64 | 88.24 | 60.24 |
| DASum(L) | **44.55** | **21.42** | **41.33** | **84.16** | **90.43** | **82.92** |
| XSUM | | | | | | |
| BertSum | 38.76 | 16.33 | 31.15 | 28.70 | 61.38 | 23.56 |
| Unilm | 42.14 | 19.53 | 34.13 | 30.54 | 64.32 | 22.54 |
| FASum | 30.28 | 10.03 | 23.76 | 12.66 | 65.74 | 26.20 |
| FASum$_{FC}$ | 30.20 | 9.97 | 23.68 | 12.03 | 67.13 | 26.09 |
| BART | **45.52** | **22.48** | **37.29** | 34.83 | 65.45 | 22.98 |
| CONSEQ | 44.67 | 21.66 | 36.47 | - | - | 22.42 |
| CLIFF | - | - | 36.19 | - | - | 25.47 |
| FACTEDIT | 33.58 | 14.68 | 26.71 | 20.13 | - | 23.91 |
| LASum | 44.59 | 21.48 | 36.17 | 39.02 | 68.10 | 26.33 |
| DASum | 44.89 | 21.54 | 36.25 | 37.62 | 67.80 | 27.31 |
| DASum(L) | 44.49 | 21.06 | 35.74 | **39.74** | **69.15** | **27.43** |

Table 1: ROUGE and factual evaluation scores on CNN/DM and XSUM, where R1, R2, RL refer to ROUGE-1, ROUGE-2, ROUGE-L, respectively. DASum takes BART as backbone, while DASum(L) takes trained LASum model as backbone.

3) **FASum** (Zhu et al., 2021) builds fact graph by relation triples and feeds the graph representation to the decoder when training. Based on FASum, Zhu et al. (2021) propose a corrector FC to modify errors in summary, represented by FASum$_{FC}$ here.

4) **BART** (Lewis et al., 2020) is an autoregressive pre-trained model with denoising strategy.

5) **CONSEQ** (Nan et al., 2021) divides the model-generated summries into positive and negative set and improves factuality by maximizing the likelihood probability of positive instances and minimizing the likelihood probability of negative instances.

6) **CLIFF** (Cao and Wang, 2021) also constructs a collection of positive and negative summaries, and utilizes comparative learning to enhance the factuality.

7) **FACTEDIT** (Balachandran et al., 2022) is

a factual error correction model for summaries, which improves the factuality by modifying summaries.

8) **LASum** (Gao et al., 2022a) employs adversarial learning to integrate the fact graph into the summarization model.

Table 1 shows the ROUGE and factual evaluation scores on two datasets. On CNN/DM, DASum has improved in both the informativeness evaluation metric ROUGE and the factual evaluation metric: DAE, SENT, and FactC.

Compared with BART, our ROUGE value of DASum has increased by 0.34 on average, while the three factual evaluation metric has increased by 6.3 on average; On XSUM, DASum achieves comparable ROUGE scores. However, in terms of factual metrics, the introduction of the factual relation discrimination task helps improve the factuality of summaries and outperforms BART in all three factual metrics.This indicates that the factual relation discrimination task is helpful for factuality-oriented abstractive summarization.

Compared with LASum, DASum(L) also improves all the factual metrics on both the CNN/DM and XSUM datasets, especially on CNN/DM. This result also indicates that the factual relation discrimination task not only can help factuality-oriented abstractive summarization, but also has the generalization ability to boost different models of factuality-oriented abstractive summarization.

CONSEQ has very similar Rouge scores to our DASum variations. The reason for the similar Rouge scores is that both CONSEQ and our DASum variations utilize Bart-large as the backbone. CONSEQ specifically employs the Fairseq implementation of BART-large as the summarization model and fine-tunes the BART-large model on CNN/DM and XSUM datasets. On the other hand, our work makes use of facebook/bart-large-cnn and facebook/bart-large-xsum as the pre-training parameter for the DASum model on the CNN/DM and XSUM datasets, respectively. Moreover, in comparison to CONSEQ, our DASum incorporates the factual relation discrimination task to enhance the factuality of abstractive summarization by addressing diverse factual errors. Hence, our DASum can improve the factuality metrics FactCC/DAE/SENT due to the factual relation discrimination and both CONSEQ and our DASum achieve the similar R1/R2/RL because both of them take Bart-large as the backbone.

| | | CNN/DM | | | XSUM | |
|---|---|---|---|---|---|---|
| Model | DAE | SENT | FactC | DAE | SENT | FactC |
| Baseline | 74.75 | 85.34 | 56.16 | 35.16 | 65.30 | 23.47 |
| +Signal Cha | 77.64 | 88.24 | 60.24 | 37.62 | 67.80 | 27.31 |
| +Signal Emb | 76.15 | 87.72 | 56.82 | 37.56 | 66.12 | 27.45 |

Table 2: The ablation results of factual signals.

The emergence of advanced instruction-tuned large language models (LLMs, e.g., ChatGPT [1]), has presented exciting possibilities for factuality-oriented abstractive summarization. Zhang et al. (2023) report that ChatGPT can achieve 28.00 (w/o knowledge extractor) and 36.00 (w/ knowledge extractor) in FactC and these figures are larger than ours. However, the informativeness metrics ROUGEs of ChatGPT are very lower than the existing fine-tuning models and ours. It only obtains 21.92, 5.98 and 17.62 in R1, R2 and RL, respectively Zhang et al. (2023). The gap between the factual metrics and informativeness metrics show ChatGPT is still an preliminary model for abstractive summarization. By contrast, our DASum can achieve both the high factual and informativeness values.

### 4.3 Ablation Study

The factual signal ablation experiments are designed to test whether the factual signal has a guiding effect on the model's generation of factual summaries. The experimental results are presented in Table 2. Baseline refers to the DASum model without factual signals, while Cha refers to the model using special characters as factual signals and Emb refers to the model using embedding vectors as guidance signals. Specifically, when adding Position Embedding, a factual label embedding should also be added to indicate whether it is factually correct or not.

Table 2 indicates that the two models, Cha and Emb, with the addition of factual signals, outperform Baseline in terms of factuality, which proves that the factual signals can guide the model to generate summaries with strong factuality. In addition, Cha outperforms Emb in factual metrics due to that Emb requires the model to learn the embedding of the label. Cha uses special characters as guiding signals, which is simpler and easier for the model to understand.

---

[1] https://chat.openai.com/chat

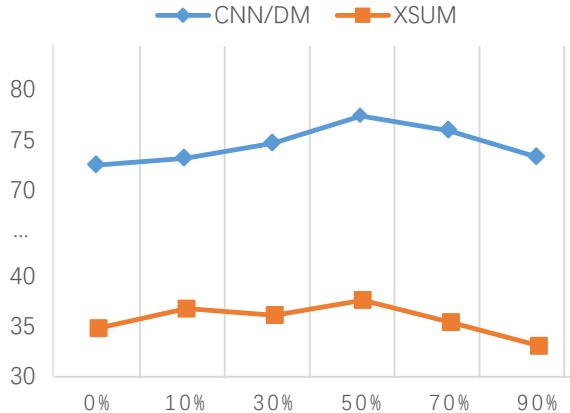

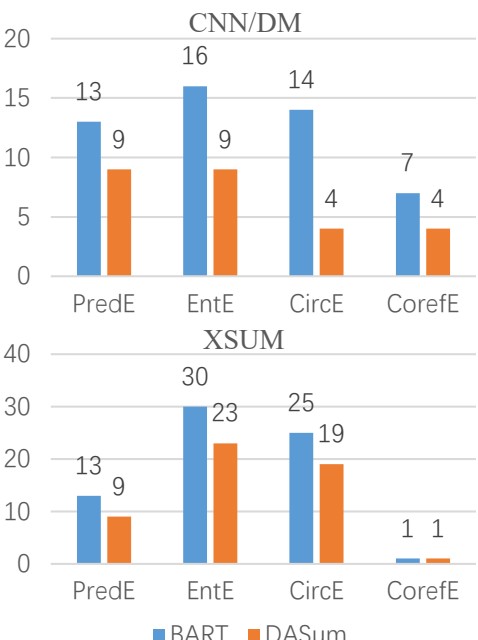

Figure 3: The DAE scores under different proportions of negative samples.

## 4.4 Proportion of Positive and Negative Instances

The comparative experiments on the proportion of positive and negative samples in the factual summarization dataset is used to verify the impact of the proportion of negative samples on model performance. The experimental results are shown in Fig. 3, where the horizontal axis represents the proportion of negative samples in the whole dataset, and the vertical axis represents the score of the DAE (the experiments on the other factual metrics show the similar results).

We can find that as the proportion of negative samples in the factual summarization dataset increases, the DAE score tends to increase and then decrease. For the CNN/DM and XSUM datasets, the factual summarization dataset with a negative sample proportion of 50% has the highest DAE score. When the negative proportion is less than 50%, the factuality of the model gradually improves, indicating that the construction of negative samples is beneficial for the model to enhance its understanding of facts; When the proportion of negative samples exceeds 50%, the factuality of the model shows a downward trend. This is mainly because when there are too many negative samples, the factual guidance signal tends to be biased towards the negative sample side, and the guidance effect on the positive sample side becomes weaker.

## 4.5 Human Evaluation on Different Errors

DASum enhances the factuality of summaries through factual relation discrimination. To further verify the effectiveness of factual relation discrimination, we manually evaluate the summaries gener-

Figure 4: The number of instances with different categories of factual errors.

ated by DASum and BART.

Consistent with Gao et al. (2022a), the human evaluation is also conducted by three experts with NLP experience, who independently provide factual labels for each instance according to the category of factual errors defined by Pagnoni et al. (2021). Pagnoni et al. (2021) find that 69% of factual errors are Semantic Frame errors (PredE, EntE, CircE) and Coreference error (CorefE). Semantic Frame errors mean the semantic frame of the generated summary sentence is wrong. Semantic frame consists of predicates and frame elements, which can be divided into core and non-core frame elements. The former is crucial for the meaning of the semantic frame, such as subject and object, while the latter yields additional descriptive information, such as location and date. Predicate Error (PredE), Entity Error (EntE) and Circumstance Error (CircE) indicates factual errors of predicates, core and non-core frame elements, respectively. Coreference error interprets pronouns and other types of references to previously mentioned entities that are either incorrect or have no explicit antecedents, making them disputable. Specifically, the evaluator will first assess the factuality of the entire summary. If marked as factually incorrect, they will determine the specific category of factual error.

We randomly select 100 samples from the CNN/DM and XSUM test sets for human eval-

| |
|---|
| **Article-1:** London Welsh have announced a contract extension for their former England international back Olly Barkley. The 33-year-old joined the Exiles last year, having previously played for Bath, Gloucester, Racing Metro, Grenoble and the Scarlets. He won 23 Test caps, the last of which was against New Zealand in 2008. ··· |
| **Target Summary:** Olly Barkley has signed a contract extension at London Welsh. The Exiles were relegated from the Aviva Premiership this season. Fly half Barkley is keen to fire the Welsh back up to the top flight. CLICK HERE for all the latest rugby unions news. |
| **BART:** Olly Barkley has extended his stay at London Welsh. ··· Barkley won 23 England caps, the last of which was in 2008. |
| **DASum:** Olly Barkley has extended his stay at London Welsh. The 33-year-old has previously played for Bath, Gloucester, Racing Metro, Grenoble and the Scarlets. Welsh are set to return to the Championship next season after finishing bottom of the Aviva Premiership. |

Table 3: Example of human evaluation.

uation. For CNN/DM, there are 30 instances with factual errors for BART, while 17 for DASum; For XSUM, there are 46 instances with factual errors for BART, while 37 for DASum.

Fig 4 shows the different categories of factual errors contained in the summaries generated by BART and DASum, as well as their corresponding numbers. We can see that DASum significantly alleviates semantic framework errors and coreference errors. For example, on CNN/DM, the number of EntE decreases from 16 (BART) to 9 (DASum); On XSUM, the number of EntE decreased from 30 (BART) to 23 (DASum).

Table 3 shows an example of human evaluation, in which the generated summary of BART has entity error. It confuses "Test caps" with "England caps", while our DASum model generates the summary with the right dependency relations.

## 5 Conclusion

In this paper, we employ the factual relation discrimination task to deepen the model's understanding of factual errors and guide it to generate factual summaries. Specifically, we construct a factual summarization dataset with positive and negative samples through data augmentation. Then, we train the model to identify factual samples, discover the locations and reasons for factual errors, and guide the model to generate summaries through factual guidance signals. Experimental results on CNN/DM and XSUM show that our DASum outperforms the baselines on factual metrics. Our future work will focus on how to construct more accurate negative samples to further identify the fine-grained causes of factual errors.

## Limitations

Although our proposed DASum model has made strides in improving factual accuracy, there remains a disparity between its information and that of state-of-the-art models. This presents an area for future improvement. Additionally, in subsection 3.1, we strive to simulate the distribution of factual errors to artificially construct those with factual errors. However, these constrained construction strategies cannot always encompass all types of errors. Therefore, our proposed model cannot address all types of factual errors in summaries.

Furthermore, solely relying on our factual relation discrimination task is insufficient to effectively address all factual inconsistency issues. It may be beneficial to incorporate additional discrimination tasks in order to enhance performance. Additionally, each assessment metric pertaining to abstractive summarization possesses strengths and weaknesses. Thus, it is important to incorporate a wide range of automated evaluation metrics (e.g. SummAC for entailment and FactEval for QA) or propose more inclusive evaluation metrics.

## Acknowledgements

The authors would like to thank the three anonymous reviewers for their comments on this paper. This research was supported by the National Natural Science Foundation of China (Nos. 62276177, 61836007, and 62376181), and Project Funded by the Priority Academic Program Development of Jiangsu Higher Education Institutions (PAPD).

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
