# OpenReview forum: "Factual Relation Discrimination for Factuality-oriented Abstractive Summarization"
_EMNLP/2023/Conference — EMNLP 2023 Findings_

### Official Review · Reviewer_FTBC · 2023-07-24

**Typos Grammar Style And Presentation Improvements:** There are some typos. Please proof-re…
**Soundness:** 3

**Excitement:**

3: Ambivalent: It has merits (e.g., it reports state-of-the-art results, the idea is nice), but there are key weaknesses (e.g., it describes incremental work), and it can significantly benefit from another round of revision. However, I won't object to accepting it if my co-reviewers champion it.

**Missing References:**

None that I can see.

**Paper Topic And Main Contributions:**

In order to identify the root causes of factual errors, the paper suggests an abstractive summarization model that is factuality-oriented and is based on factual relation discrimination. The authors provide negative examples of summaries using data augmentation techniques and create a dataset for factual summarizing. Following that, students complete the factual relation discrimination task, which assesses the veracity of the dependencies in summaries during summary generation, enhancing the factuality of summaries.

**Questions For The Authors:**

L.073 mtrics => metrics

Table 1, part 1: CONSEQ has very similar Rouge scores to DASum variations. Is there an explanation for this, maybe an intuitive one? What are the similarities between the two methods that cause this phenomenon?

**Reasons To Accept:**

The idea is nice and well described, and the results are slightly above current state-of-the-art.

**Reasons To Reject:**

Some explanation about the results would be nice - no intuition of why the method works better is provided.

**Reproducibility:**

4: Could mostly reproduce the results, but there may be some variation because of sample variance or minor variations in their interpretation of the protocol or method.

**Reviewer Confidence:**

4: Quite sure. I tried to check the important points carefully. It's unlikely, though conceivable, that I missed something that should affect my ratings.

---

> ### Author Rebuttal · Authors · 2023-08-27
>
> Thank you very much for your valuable comments.
>
> **Q1: Some explanation about the results would be nice - no intuition of why the method works better is provided.**
>
> **A1**: In this paper, we use data augmentation methods to introduce negative samples to the task of abstractive summarization. Specifically, we use five operations (i.e., pronoun swapping, sentence negation, time and date entity swapping, quantifier swapping, and named entity swapping) to construct counterfactual summaries. These negative samples with the error causes (i.e., the above five operations) can help the factual relation discrimination to identify counterfactual relations and then guide the abstractive summarization model to generate factual summaries.
>
> In the section “Human Evaluation on Different Errors”, we randomly select 100 samples from the CNN/DM and XSUM test sets for human evaluation. For CNN/DM, BART has 30 instances of factual errors while DASum has 17; for XSUM, BART has 46 instances of factual errors while DASum has 37. As shown in Fig.4, DASum can considerably reduce Predicate Error (PredE), Entity Error (EntE), and Circumstance Error (CircE). These results confirm that the factual relation discrimination is effective for abstractive summarization.
>
> **Q2: Table 1, part 1: CONSEQ has very similar Rouge scores to DASum variations. Is there an explanation for this, maybe an intuitive one? What are the similarities between the two methods that cause this phenomenon?**
>
> **A2**: The reason for the similar Rouge scores is that CONSEQ and our DASum both utilize Bart-large as the backbone. CONSEQ specifically employs the Fairseq implementation of BART-large as the summarization model and fine-tunes the BART-large model on CNN/DM and XSUM datasets. On the other hand, our work makes use of facebook/bart-large-cnn and facebook/bart-large-xsum as the pre-training parameter for the DASum model on the CNN/DM and XSUM datasets, respectively. Moreover, in comparison to CONSEQ, our DASum incorporates the factual relation discrimination task to enhance the factuality of abstractive summarization by addressing diverse factual errors. Hence, our DASum can improve the factuality metrics FactCC/DAE/SENT due to the factual relation discrimination and both CONSEQ and our DASum achieve the similar R1/R2/RL because both of them take Bart-large as the backbone.

---

### Official Review · Reviewer_QpAJ · 2023-08-04

**Soundness:** 4

**Excitement:**

3: Ambivalent: It has merits (e.g., it reports state-of-the-art results, the idea is nice), but there are key weaknesses (e.g., it describes incremental work), and it can significantly benefit from another round of revision. However, I won't object to accepting it if my co-reviewers champion it.

**Paper Topic And Main Contributions:**

This paper proposes a new method for factuality oriented abstractive summarization.


**Reasons To Accept:**

The problem of factuality is a practical concern of abstractive summarization systems and this work provides a novel approach to improve the factuality of abstractive summarization systems. This approach can be easily incorporated into new autoregressive models. The evaluation is thorough (both automatic and human).


**Reasons To Reject:**

- It would be important to include more diverse automated evaluation metrics like SummAC (entailment) or FactEval (QA). DAE and FactCC might are similar in spirit to the training paradigm in this paper and there could be confounding factors.
- The method chosen to obtain feature representations of the relations seems overly complex and no ablation is reported.
- Like in other data augmentations methods introducing factual errors in summaries (ex: FactCC), there are limitations in the type of errors that can be modeled. These tend to lack the distribution of naturally occurring errors.
- The improvements in terms of factuality are slightly lower than LASum.
- Lacking a more thorough limitations section.

**Reproducibility:**

3: Could reproduce the results with some difficulty. The settings of parameters are underspecified or subjectively determined; the training/evaluation data are not widely available.

**Reviewer Confidence:**

3: Pretty sure, but there's a chance I missed something. Although I have a good feel for this area in general, I did not carefully check the paper's details, e.g., the math, experimental design, or novelty.

**Typos Grammar Style And Presentation Improvements:**

Please restructure the section where you present the types of errors introduced as the format does not make it easy to understand which types of errors are considered. The term "swapped" is also confusing.

Please ensure the paper goes through grammar and spell checking.

---

> ### Author Rebuttal · Authors · 2023-08-27
>
> Thank you very much for your valuable comments.
>
> **Q1: It would be important to include more diverse automated evaluation metrics like SummAC (entailment) or FactEval (QA).**
>
> **A1**: Yes. A wider variety of automated evaluation metrics can provide a more comprehensive evaluation of the proposed model. However, most previous studies reported FactCC, DAE, and SENT, while a few reported only FactCC (e.g., CONSEQ, CLIFF, and FACTEDIT). For comparison with previous studies, we report the above three metrics.
>
> **Q2: The method chosen to obtain feature representations of the relations seems overly complex and no ablation is reported.**
>
> **A2**: In Subsection 3.2, only the boundary embedding and length embedding are introduced for representing factual relations. BART is used to produce the boundary embedding, which is superior to other options such as BERT and RoBERTa. The length embedding is determined by the length of dependent and core words. Our experimental results indicate that the enhancements primarily arise from the boundary embedding. Using BERT/RoBERTa/XLNET instead of BART results in significant drops in various metrics, including FactCC, DAE, and SENT (by about 2.0-5.0). In addition, removing the length embedding results in a decrease of 0.3 for FactCC, DAE, and a decrease of 1.4 for SENT.
>
> **Q3: Lacking a more thorough limitations section.**
>
> **A3**: We revised this section as follows:
>
> Although our proposed DASum model has made strides in improving factual accuracy, there remains a disparity between its information and that of state-of-the-art models. This presents an area for future improvement. Additionally, in subsection 3.1, we strive to simulate the distribution of factual errors to artificially construct those with factual errors. However, these constrained construction strategies cannot always encompass all types of errors. Therefore, our proposed model cannot address all types of factual errors in summaries. Furthermore, solely relying on our factual relation discrimination task is insufficient to effectively address all factual inconsistency issues. It may be beneficial to incorporate additional discrimination tasks in order to enhance performance. Additionally, each assessment metric pertaining to abstractive summarization possesses strengths and weaknesses. Thus, it is important to incorporate a wide range of automated evaluation metrics (e.g. SummAC for entailment and FactEval for QA) or propose more inclusive evaluation metrics.
>
> **Q4: The improvements in terms of factuality are slightly lower than LASum.**
>
> **A4**: We have evaluated the effectiveness of our proposed model, DASum and DASum(L), using BART and LASum as backbones. DASum outperforms BART, improving FactCC, DAE, and SENT by 10.64, 4.89, and 3.30 on CNN/DM and 4.33, 2.79, and 2.35 on XSUM, respectively. Additionally, DASum(L) outperforms LASum, improving FactCC, DAE, and SENT by 0.52, 1.65, and 2.82 on CNN/DM and 1.10, 0.72, and 1.05 on XSUM, respectively.
>
> **Q5: Like in other data augmentations methods introducing factual errors in summaries (ex: FactCC), there are limitations in the type of errors that can be modeled. These tend to lack the distribution of naturally occurring errors.**
>
> **A5**: There are numerous categories of factual errors present in abstractive summaries, including entity errors, entity relation errors, predicate errors, coreference errors, and commonsense errors. It would be advantageous for a model to address all of these types of errors, rather than focusing solely on a few. Presently, nearly all current methods concentrate on only a limited number of factual errors.
> As explained in Appendix A, Pagnoni et al. (2021) classify factual errors and find that 69% of these errors consist of predicate, entity, circumstance, and coreference errors. Therefore, we present the factual relation discrimination task to tackle five types of factual errors: pronoun errors, sentence negation errors, time and date errors, quantifier errors, and entity errors. This improves the accuracy of abstractive summaries.

---

### Official Review · Reviewer_FEBY · 2023-08-05

**Soundness:** 3

**Excitement:**

3: Ambivalent: It has merits (e.g., it reports state-of-the-art results, the idea is nice), but there are key weaknesses (e.g., it describes incremental work), and it can significantly benefit from another round of revision. However, I won't object to accepting it if my co-reviewers champion it.

**Paper Topic And Main Contributions:**

This paper proposes a factuality-oriented abstractive summarization model based on a new task factual relation discrimination to mine the causes of factual errors. A factual summarization dataset is first built, and then a factual relation discrimination task is proposed to improve the factuality of dependency relations in summaries. Experiments demonstrate the effectiveness of the proposed method.

**Reasons To Accept:**

- The proposed method is clearly explained and is easy to follow.
- Comprehensive experiments and significant improvements compared with other counterparts.

**Reasons To Reject:**

- I think the novelty of constructing factual summarization dataset seems a little limited. Similar construction methods have already been proposed in other factual summarization works (e.g., CLIFF). What are the main differences compared with those works?
- I think the designed factual relation discrimination task alone is not enough to solve the factual inconsistency problem well, and the factuality problem in summarization not only contains the incorrect factual relations. More discrimination tasks can be introduced to further improve the performance.

**Reproducibility:**

3: Could reproduce the results with some difficulty. The settings of parameters are underspecified or subjectively determined; the training/evaluation data are not widely available.

**Reviewer Confidence:**

3: Pretty sure, but there's a chance I missed something. Although I have a good feel for this area in general, I did not carefully check the paper's details, e.g., the math, experimental design, or novelty.

---

> ### Author Rebuttal · Authors · 2023-08-27
>
> Thank you very much for your valuable comments.
>
> **Q1: What are the main differences compared with CLIFF (2021)?**
>
> **A1**: The primary differences between CLIFF and our work stem from two aspects. First, while CLIFF only takes into account entity substitution when building factual summarization datasets, we not only consider entity substitution, but also carry out pronoun substitution to address coreference errors and perform sentence negation to enhance the model's comprehension of sentence polarities. Additionally, both CLIFF and our work employ the contrastive approach. The distinction lies in the fact that while CLIFF aims to maximize the cosine similarity of positive pairs, our approach assigns higher probabilities to positive samples and lower probabilities to negative samples, which is more intuitive and facilitates model learning. Finally, the human evaluation on different errors shows that our model DASum can reduce coreference, circumstance and predicate errors, in comparison with CLIFF.
>
> **Q2: More discrimination tasks can be introduced to further improve the performance.**
>
> **A2**: Yes. There are several kinds of factual errors found in abstractive summaries, such as entity errors, entity relation errors, predicate errors, coreference errors, and commonsense errors. To improve the factuality of abstractive summarization, it is recommended to include additional discrimination tasks. Currently, most existing methods concentrate on only a few kinds of factual errors. In our paper, we present the factual relation discrimination task to tackle five types of factual errors, including pronoun errors, sentence negation errors, time and date errors, quantifier errors, and entity errors.
>
> Besides, the term "relation" in "factual relation discrimination" refers to the use of the dependency relation to model the aforementioned factual errors, rather than solely focusing on relation errors.

---

### Meta-Review · Area_Chair_584a · 2023-09-16

**Recommendation:** 4

**Metareview:**

The paper proposes a factuality-oriented abstractive summarization model DASum. The model is based on the factual relation discrimination task, guiding DASum to generate true factual relations. Also, the authors construct a new factual summarization dataset using data augmentation to generate negative samples. Experimental results on two datasets demonstrate the DASum's superiority over several SOTA benchmarks on factual metrics.

The reviewers mentioned several pros of this work, such as:
1. The proposed method is novel,
2. The method is clearly explained and is easy to follow,
3. Comprehensive experiments show significant improvements compared with other counterparts.
4. The evaluation is thorough and contains both automatic and human parts.

I also want to mention the contribution to the data resources for the task of factual summarization.

The authors addressed all comments (including missing evaluation metrics, motivating the choice of augmentation methods, lacking a more thorough limitations section, etc.) in their rebuttal.

---

### Decision · Program_Chairs · 2023-10-07

**Decision:**

Accept-Findings

**Comment:**

The paper proposes a factuality-oriented abstractive summarization model DASum. The model is based on the factual relation discrimination task, guiding DASum to generate true factual relations. Also, the authors construct a new factual summarization dataset using data augmentation to generate negative samples. Experimental results on two datasets demonstrate the DASum's superiority over several SOTA benchmarks on factual metrics.

The reviewers mentioned several pros of this work, such as:
1. The proposed method is novel,
2. The method is clearly explained and is easy to follow,
3. Comprehensive experiments show significant improvements compared with other counterparts.
4. The evaluation is thorough and contains both automatic and human parts.

I also want to mention the contribution to the data resources for the task of factual summarization.

The authors addressed all comments (including missing evaluation metrics, motivating the choice of augmentation methods, lacking a more thorough limitations section, etc.) in their rebuttal.